# Study of Fractal Dimensions of Microcrystalline Cellulose Obtained by the Spray-Drying Method

## Michael Ioelovich

Celdesigner Ltd, 2 Bergman St., Rehovot 7670504, Israel; ioelovichm@gmail.com; Tel.: +972-0893-66612

**Abstract:** In this research, the fractal structure of beads of different sizes obtained by the spray-drying of aqueous dispersions of microcrystalline cellulose (MCC) was studied. These beads were formed as a result of the aggregation of rod-shaped cellulose nanocrystalline particles (CNP). It was found that increasing the average radius (R) of the formed MCC beads resulted in increased specific pore volume (P) and reduced apparent density ($\rho$). The dependences of P and $\rho$ on the scale factor (R/r) can be expressed by power-law equations: $P = P_o (R/r)^{E-D_p}$ and $\rho = d (R/r)^{D_d - E}$, where the fractal dimensions $D_p = 2.887$ and $D_d = 2.986$ are close to the Euclidean dimension E = 3 for three-dimensional space; r = 3 nm is the radius of the cellulose nanocrystalline particles, $P_o = 0.03$ cm$^3$/g is the specific pore volume, and d = 1.585 g/cm$^3$ is the true density (specific gravity) of the CNP, respectively. With the increase in the size of the formed MCC beads, the order in the packing of the beads was distorted, conforming to theory of the diffusion-limited aggregation process.

**Keywords:** microcrystalline cellulose (MCC); spray-drying; MCC beads; specific pore volume; apparent density; fractal dimension

## 1. Introduction

It is known that various natural and artificial objects and phenomena can be considered as fractals, distinctive features of which are scale invariance (self-similarity) and fractional dimension [1,2]. The theory of fractals is widely used in engineering, mathematics, biology, physics, chemistry, and other areas. According to this theory, the fractal dimension (D) of an object can be determined by logarithmization of power-law dependence of a structure or property on the scale factor. In particular, the theory of fractals was applied to describe the structure and properties of such wide-spread natural biopolymers as cellulose as well as diverse cellulose materials.

For instance, cellulose fibers were studied by a method of low-temperature nitrogen sorption to measure the dependence of the cumulative volume on the radius of various pores expressed by the power-law function, from which the fractal dimension from 2.88 to 2.95 was determined [3]. In another study [4], the fractal structure of pores in various cellulose materials was studied by nitrogen and water vapor sorption methods; in the case of nitrogen sorption the fractal dimension of pores was from 2.13 to 2.50, whereas the sorption of water vapor gave the fractal dimension of pores less than 1.5 due to the alteration of the cellulose structure under the effect of water. The study of the distribution of cellulose aggregates by the small-angle X-ray scattering method permitted the calculation of the fractal dimension D = 2.10 [5].

An important cellulose product is microcrystalline cellulose (MCC) with an increased crystallinity, which is obtained by the depolymerization of cellulose materials with dilute mineral acids at increased temperatures up to the level-off degree of polymerization (LODP) from 120 to 250 [6,7]. MCC is used as an excipient for tablets, cosmetic formulation, and food products, as well as a filler for various composite materials and a special additive for some technical applications. To produce medical and food grade MCC, cellulose feedstock (bleached wood pulp or purified cotton cellulose) is hydrolyzed

with 1–3 M mineral acids at increased temperatures to LODP, followed by washing to neutral pH, dilution with water, and spray-drying of the dispersion to obtain dry MCC beads [8].

The main purpose of this research was to study the fractal characteristics of MCC beads with different sizes prepared by the spray-drying method.

## 2. Materials and Methods

The initial material was chemical-grade cotton cellulose (99% α-cellulose, DP = 2700) obtained from Hercules Inc.

Cotton cellulose was hydrolyzed with boiling 1.5 M sulfuric acid at the acid/cellulose ratio of 10 for 1 h followed by the filtration of the acid and the washing of MCC on the filter until a neutral pH was obtained. The resulting wet cake of MCC was diluted with distilled water to obtain 1–5% dispersions, which were disintegrated in a Waring blender at 15,000 rpm for 10 min to break the agglomerates. To produce the beads, aqueous dispersions of MCC were spray-dried using a lab drier by Pilotech at the following conditions: feeding 10 mL/min, air pressure 0.2 MPa, inlet temperature 120 °C, and outlet temperature 60 °C.

MCC beads of different sizes were separated by screening through sieves with mesh sizes of 80–100, 140–170, 230–270, and 450–635. The average radii of the beads are shown in Table 1.

**Table 1.** Average radii (R) of the separated MCC beads.

| Mesh | Hole Diameter, μm | R, μm |
|---|---|---|
| 80–100 | 149–177 | 82 |
| 140–170 | 88–105 | 48 |
| 230–270 | 53–63 | 29 |
| 450–635 | 20–32 | 13 |

The size and shape of the MCC beads were investigated by scanning electron microscopy (Hitachi S-4700). With rise of the concentration of MCC dispersion from 1 to 5%, an increase in the yield of larger beads was observed.

The nanostructure of MCC beads was investigated by the wide-angle X-ray scattering (WAXS) method using a Rigaku-Ultima Plus diffractometer (CuK$_\alpha$ radiation, $\lambda$ = 0.15418 nm) [9,10]. The lateral size (L) of the cellulose nanocrystalline particles (CNP) in the direction perpendicular to the [200] planes of the crystalline unit cells were calculated by a modified Scherrer equation, taking into account the contribution of the instrumental factor and lattice distortions to the width of the crystalline peak. The minimum radius of the CNP was calculated as follows: r = 0.5 L.

The sorption of hexane vapor by MCC beads was measured at 25 °C with the use of a vacuum Mac-Ben apparatus with helical spring quartz scales [11]. The specific pore volume (P, cm$^3$/g) of the beads was calculated by the following equation:

$$P = V/m \qquad (1)$$

where V is total volume of pores (cm$^3$) measured at relative vapor pressure p/p$_o$ = 0.98; m is the mass of the dry sample (g).

## 3. Results

Scanning electron microscopy (SEM) studies showed that the spray-dried MCC beads had ellipsoidal or spherical shapes and contained small rod-like particles (Figure 1).

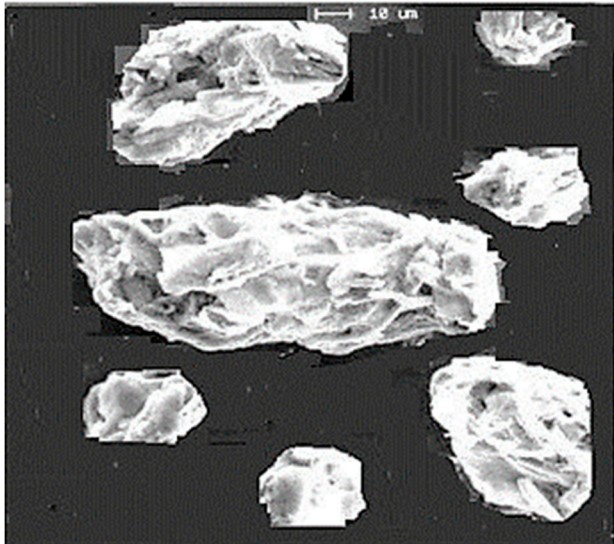

**Figure 1.** SEM image of microcrystalline cellulose (MCC) beads of different sizes.

As known, cellulose is a semicrystalline linear polysaccharide that consists of elementary nanofibrils and their bundles, called microfibrils [12]. Furthermore, each nanoscale fibril is built of ordered rod-like nanocrystallites and low ordered amorphous nanodomains. The three-dimensional ordered nanocrystallites are strong and inaccessible structural elements. In contrast, the low-ordered amorphous domains are weak and accessible segments of the fibrils. Therefore, the cleavage of amorphous domains by acid hydrolysis leads to the formation of rod-shaped nanocrystalline particles (CNP). As a result of spray-drying, the rod-shaped CNP are aggregated by their side surfaces and form micron-sized beads of microcrystalline cellulose with various average radii.

The structural studies showed that the CNP isolated from plant biomass have lateral sizes of 4–8 nm and lengths of 100–200 nm [13]. As it follows from the WAXS results, the average lateral size (L) of the CNP made of cotton cellulose is 6 nm and their average radius (r) is 3 nm or 0.003 μm (Table 2).

**Table 2.** Lateral size (L) and minimum radius (r) of nanocrystalline particles (CNP) in MCC beads of different sizes.

| R, μm | L, nm | r, nm |
|---|---|---|
| 82 | 6.2 | 3.1 |
| 48 | 5.8 | 2.9 |
| 29 | 6.0 | 3.0 |
| 13 | 6.1 | 3.0 |
| **Average** | **6.0** | **3.0** |

The study of the vapor sorption of an inert organic liquid (hexane) revealed that the specific pore volume of MCC beads varies in the range from 0.0757 to 0.0931 $cm^3$/g (Table 3). Moreover, when the average size of the beads increases, their porosity rises.

**Table 3.** Specific pore volume (P) and apparent density (ρ) of MCC beads of different sizes.

| R, μm | R/r | P, $cm^3$/g | ρ, g/$cm^3$ |
|---|---|---|---|
| 82 | 6.2 | 3.1 | 1.378 |
| 48 | 5.8 | 2.9 | 1.390 |
| 29 | 6.0 | 3.0 | 1.400 |
| 13 | 6.1 | 3.0 | 1.415 |

The dependence of the specific pore volume (P) on the scale factor (R/r) can be expressed by the power-law function:

$$P = P_o \, (R/r)^{E-D_P} \tag{2}$$

where $P_o$ is the specific pore volume of the CNP, $D_P$ is the fractal dimension, and E = 3 is the Euclidean dimension for three-dimensional space.

After the logarithmization of the function (2), a linear graph was drawn (Figure 2), from which the values of $P_o$ = 0.03 (cm$^3$/g) and $D_p$ = 2.887 were found.

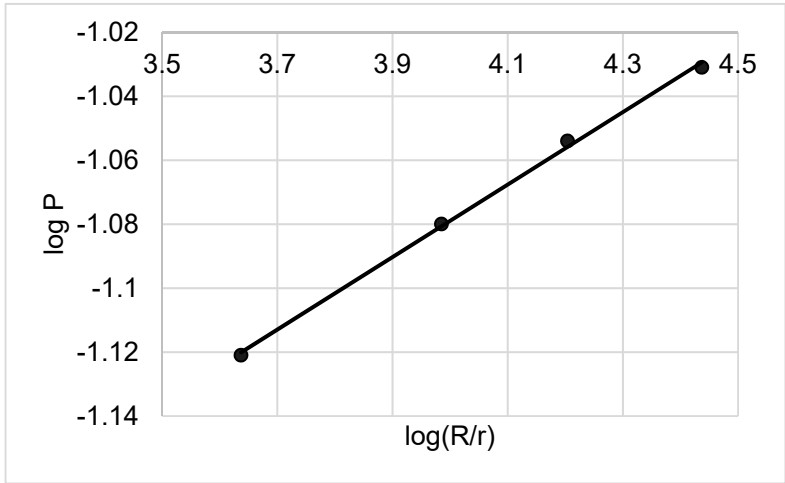

**Figure 2.** Linearized dependence P = f(R/r) in double logarithmic coordinates.

Along with the specific pore volume, it is also possible to calculate the apparent density of MCC beads, as follows:

$$\rho = (P + V_c)^{-1} \tag{3}$$

where $V_c = d^{-1}$ is the specific volume of the CNP with a crystallinity of about 80% and a true density (specific gravity) of d = 1.585 g/cm$^3$ [12].

Using $\rho$ values and the scale factor (R/r) (Table 3), the linear dependence can be obtained (Figure 3) by the logarithmization of the function:

$$\rho = d \, (R/r)^{D_d - E} \tag{4}$$

From the linearized graph (Figure 3), the value of $D_d$ = 2.986 was calculated.

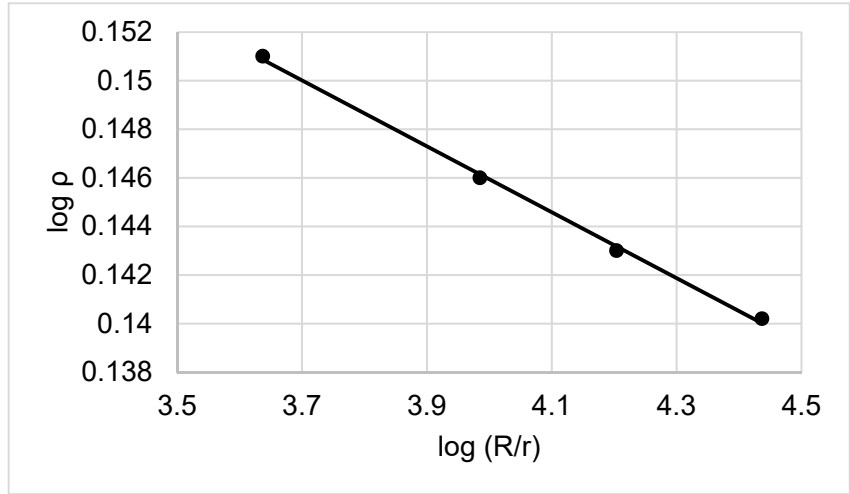

**Figure 3.** Linearized dependence $\rho$ = f(R/r) in double logarithmic coordinates.

## 4. Discussion

The rod-shaped crystalline nanoparticles (CNP) of cotton cellulose with a radius of r = 3 nm have a compact packing with a high true density (specific gravity) of d = 1.585 g/cm$^3$ and a negligible specific pore volume of $P_o$ = 0.03 cm$^3$/g. During spray-drying, a lateral aggregation of rod-shaped CNP occurs. Moreover, with the development of the aggregation process of the CNP and the increase in size of the formed MCC beads, the order in the packing is distorted, conforming to theory of the diffusion-limited aggregation process [2,14–16]. The consequence of this phenomenon is an increase in the specific pore volume (P) and a decrease in the apparent density (ρ) with the increase in size (average radius R) of the beads. It was found that the dependences of P and ρ on the scale factor (R/r) can be expressed by the power-law equations:

$$P = P_o \, (R/r)^{E-D_P} \text{ and } \rho = d \, (R/r)^{D_d \, -E}$$

where the fractal dimensions $D_P$ = 2.887 and $D_d$ = 2.986.

## 5. Conclusions

Fractal dimensions of MCC beads prepared by the spray-drying method were studied. The fractal dimensions $D_P$ = 2.887 for the specific pore volume (P) and $D_d$ = 2.986 for the apparent density (ρ) of MCC beads are close to the Euclidean dimension E = 3 for three-dimensional space. Since MCC beads are fractal objects, their characteristics such as P and ρ are power functions of the scale factor; namely, with the increase of the average radius of MCC beads the specific pore volume increases, whereas the apparent density decreases. Knowledge of these fractal features is important for obtaining MCC with desired properties.

**Funding:** This research received no external funding.

**Conflicts of Interest:** The author declares no conflict of interest.

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
