# Peer review of "Study of Fractal Dimensions of Microcrystalline Cellulose Obtained by the Spray-Drying Method"

_fractalfract, doi:10.3390/fractalfract3010003_

Reviewer 1 Report

20181220-fractalfract-413649-peer-review-v1

Study of fractal dimensions of microcrystalline 2 cellulose obtained by method of spray drying.

The meaning of the term “reduced density” should be changed (or named together) to apparent density, i.e. (mass)/(pore volume+solid volume).

The term “porosity” usually refers to the void fraction of a solid. The parameter P in the article is rather a specific pore volume, not a porosity.

The meaning and usefulness of the fractal dimensions calculated should be clearly indicated.

Despite the title of the paper and the main purpose indicated in the introduction, no references to fractal characteristics are made in the conclusions section.

Publication in the present form is not recommended.

Author Response

Thank you for the important comments. I agree with them. Therefore, the manuscript was revised:

(1). The term “porosity” was replaced with “specific pore volume”, and the term “density” was replaced with “apparent density”.

(2). The Conclusions were rewritten: the fractal characteristics of MCC beads were included; furthermore, usefulness of the calculated fractal dimensions for MCC was explained.

(3). In addition, some corrections were made to Introduction and Discussion.

Reviewer 2 Report

Page 3, line 77: SEM means scanning electron microscope?

Page 3, line 94-95: In Table 2,  for L, (6.2+5.8+6.0+6.1)/4=6?

Author Response

(1). Page 3, line 77: SEM means scanning electron microscope? Answer: Yes, SEM is scanning electron microscopic; this term was explained in the revised manuscript.

(2). Page 3, line 94-95: In Table 2, for L, (6.2+5.8+6.0+6.1)/4=6? Answer: Result is 6.025, i.e. circa 6.0.

Reviewer 3 Report

În my opinion, the paper under review can be accepted after some minor improvements

Author Response

Thank you very much for review of my article. It was revised and improved.

(1). The terms “porosity” and “density” was replaced.

(2). The Conclusions were rewritten: the fractal characteristics of MCC beads were included; furthermore, usefulness of the calculated fractal dimensions for MCC was explained.

(3). In addition, some corrections were made to Introduction and Discussion